# Fahr’s Syndrome with Pseudohypoparathyroidism: Oral Features and Genetic Insights

**DOI:** 10.3390/ijms252111611

**Published:** 2024-10-29

**Authors:** Xiangpu Wang, Taoyun Xu, Yulong Zhu, Xiaohong Duan

**Affiliations:** State Key Laboratory of Oral & Maxillofacial Reconstruction and Regeneration, National Clinical Research Center for Oral Disease, Shaanxi Key Laboratory of Stomatology, Department of Oral Biology & Clinic of Oral Rare Diseases and Genetic Diseases, School of Stomatology, The Fourth Military Medical University, Xi’an 710000, China; wxp901120@163.com (X.W.); 18575923055@163.com (T.X.); yj598968493@163.com (Y.Z.)

**Keywords:** pseudohypoparathyroidism, Fahr’s syndrome, oral clinical features

## Abstract

Fahr’s syndrome is a rare neurodegenerative disorder with limited research on its oral manifestations. This study investigates the dental features and genetic background of Fahr’s syndrome through a pedigree analysis and a retrospective literature study. A clinical examination and whole-exome sequencing (WES) were conducted on a female patient with Fahr’s syndrome and pseudohypoparathyroidism, along with her family members. The patient presented with super-numerary teeth, tooth agenesis, enamel hypoplasia, and abnormal tooth eruption. The WES did not reveal any known pathogenic mutations related to pseudohypoparathyroidism or Fahr’s disease. However, genetic variations in *KIF1A*, *FZD8*, and *PDGFA* may underlie these dental abnormalities. Additionally, a retrospective analysis of 22 reported cases from PubMed and the Human Gene Mutation Database (1 January 1965–30 June 2024) was conducted with keywords such as “Fahr’s disease”, “Fahr’s syndrome”, “dental”, and “hypoparathyroidism”. The analysis showed that patients with Fahr’s syndrome, pseudohypoparathyroidism, and idiopathic hypoparathyroidism exhibited similar oral abnormalities, including tooth agenesis, root dysplasia, dental malformations, and abnormal tooth eruption. Variations in the incidence of tooth agenesis and dental malformation among these groups may be linked to differences in parathyroid hormone metabolism. These findings suggest oral abnormalities are the key local features of Fahr’s syndrome and related parathyroid disorders.

## 1. Introduction

Fahr’s syndrome is a rare neurological disorder, with a prevalence of less than one per one million individuals [1]. It is primarily characterized by abnormal intracranial calcifications in regions such as the basal ganglia, dentate nucleus, and cerebral cortex, and it was first described by German neurologist Karl Theodor Fahr in 1930 [2]. The etiology of Fahr’s syndrome is multifaceted, with both genetic and secondary factors implicated, such as calcium and vitamin D metabolism disorders, parathyroid dysfunction, and toxin exposure [3]. One of the most common secondary causes is parathyroid-related disorders including idiopathic hypoparathyroidism (IHP), autoimmune hypoparathyroidism, pseudohypoparathyroidism (PHP), pseudopseudohypoparathyroidism (PPHP), and hyperthyroidism [4]. Besides abnormal calcifications in the brain, other phenotypes such as epilepsy, memory loss, speech disorders, and motor disorders have also been reported in patients with Fahr’s syndrome. Meanwhile, a few reported patients with Fahr’s syndrome have exhibited dental abnormalities, such as delayed tooth eruption, hypodontia, and impacted teeth [5]. It is unknown whether these phenotypes are the typical local characteristics of Fahr’s syndrome or its related parathyroid disorders. The underlying mechanisms for these oral manifestations also remain unclear. Fahr’s syndrome and related parathyroid disorders often lead to the abnormal expression of parathyroid hormone (PTH) and related genes, which probably contribute to the dental and developmental anomalies observed in these patients. For instance, one of the causative genes for IHP, *TBX1*, is known to regulate the proliferation of dental progenitor cells and craniofacial development and responds to *Fst* gene mutations, which have been linked to supernumerary teeth [6,7]. Another IHP-related gene, *PTH*, encodes the parathyroid hormone, which, along with PTHrP (parathyroid hormone-related protein), binds to the PTH1R receptor, activating downstream pathways that regulate periodontal and bone tissue development, particularly during tooth eruption [8]. Additionally, *GNAS*, the gene responsible for encoding the Gsα subunit and implicated in pseudohypoparathyroidism, has been associated with dental anomalies such as enamel hypoplasia, delayed tooth eruption, and craniofacial bone malformations [9].

It is important to note that the distinction between Fahr’s disease and Fahr’s syndrome remains a topic of ongoing debate. These terms are often used interchangeably in the literature, but some experts argue for a clearer distinction [10]. Unlike Fahr’s syndrome, Fahr’s disease should refer specifically to cases of primary or idiopathic basal ganglia calcification, in which the underlying cause is unknown, and no secondary conditions are identified. Additionally, Fahr’s disease is typically inherited in an autosomal dominant pattern with incomplete penetrance and is often age-dependent, although it can also occur through autosomal recessive inheritance or sporadically [11]. To date, six pathogenic genes have been linked to Fahr’s disease: *SLC20A2* (encoding type III sodium-dependent phosphate transporter 2), *XPR1* (a retroviral receptor involved in phosphate export), *PDGFRB* (encoding platelet-derived growth factor receptor β), and *PDGFB* (encoding the B subunit of platelet-derived growth factor). Mutations in *MYORG* and *JAM2* have also been associated with the disease. Interestingly, reports of dental anomalies related to Fahr’s disease are exceedingly rare [12].

This study aims to explore the dental features and genetic background of a patient with Fahr’s syndrome and pseudohypoparathyroidism and involved collecting samples from and performing assessments of her Chinese family. We compared the common dental manifestations of Fahr’s syndrome and related parathyroid disorders such as HPP and IHP through a retrospective analysis of published cases. We also conducted a differential diagnosis between Fahr’s syndrome and Fahr’s disease, briefly discussing their respective dental features and genetic differences. Our findings may provide potential genetic insights into the shared oral abnormalities associated with these conditions.

## 2. Result

### 2.1. Clinical Features of Proband

The results of the general physical examination indicated that the proband’s vital signs were stable, with her respiratory rate, blood pressure, body temperature, and other parameters all falling within the normal range. The oral examination revealed the persistent retention of the upper left deciduous canine and the residual root of the lower right deciduous canine. Additionally, a residual root was identified in the upper left second premolar. Enamel hypoplasia was observed in the upper right second premolar, lower right first premolar, and lower left first premolar. Moreover, the upper right lateral incisor, canine, first premolar, second molar, upper left central incisor, lateral incisor, canine, first premolar, lower left canine, lower right canine, and lower right second premolar were not detectable during the clinical examination (as depicted in Figure 1). The proband also presented with suboptimal oral hygiene, a pronounced anterior open bite, and a mild midfacial retrusion.

### 2.2. Biochemical and Radiographic Analysis

The laboratory biochemical analysis revealed that the proband’s calcium level was 1.63 mmol/L, her potassium level was 2.91 mmol/L, and her parathyroid hormone level was 412.7 ng/L, indicating a diagnosis of pseudohypoparathyroidism (Table 1). The CT scans demonstrated extensive bilateral calcifications in the basal ganglia, specifically involving the caudate nucleus, putamen, globus pallidus, thalamus, and dentate nucleus (Figure 2), which is characteristic of Fahr’s syndrome.

The CBCT revealed that the patient had impacted teeth, including the right maxillary lateral incisor, canine, first premolar, and second molar, as well as the left maxillary central incisor, lateral incisor, canine, and first premolar, and both the left and right mandibular canines and first premolars. Additionally, the right lower second molar was missing, and a supernumerary tooth was observed on the palatal side of the right maxillary canine and first premolar. The patient also exhibited obstructed root development in the left lower canine, first and second premolars, right lower canine, first premolar, second molar, and other teeth. No signs of trauma, root apex tumors, cysts, or other abnormalities were detected (Figure 3). The clinical significance of these findings in the context of Fahr’s syndrome lies in the impact of metabolic disturbances, particularly calcium and phosphate imbalances, on dental development. Previous studies have suggested that these metabolic disruptions, commonly seen in pseudohypoparathyroidism and other parathyroid hormone (PTH)-related conditions, can adversely affect the formation and eruption of teeth. The presence of multiple impacted teeth and a supernumerary tooth in this patient is likely a consequence of disrupted calcium homeostasis, which interferes with normal tooth development and eruption patterns.

Furthermore, the findings of enamel hypoplasia and root dysplasia align with known complications associated with parathyroid dysfunction and impaired mineralization, which are common in pseudohypoparathyroidism. These dental anomalies not only reflect localized disturbances in the oral cavity but are also indicative of the broader systemic effects of calcium and phosphate metabolism disorders that characterize Fahr’s syndrome.

### 2.3. Molecular Genetic Analysis

In the WES analysis of the proband and their family members, 99% of the target areas exhibited an average coverage rate of >150×, indicating an accurate and comprehensive selection of reference sequences. A total of 43,057 genetic variants were detected among the family members: 30,547 in the proband, 30,258 in her father, 30,235 in her mother, 31,382 in her eldest brother, and 31,226 in her second brother. Following the genetic inheritance pattern, 475 homozygous variants were identified as conforming to autosomal recessive inheritance. Furthermore, 148 de novo homozygous variants and 460 de novo heterozygous variants were found in autosomal genes. Variants with an MAF greater than 0.005 were initially excluded. A subsequent analysis utilizing various public databases significantly reduced the number of candidate variants by considering types, consequences, and gene duplication. Ultimately, after a careful evaluation, fifteen variants were identified. The study identified five genes (*OR2T35*, *KIF1A*, *IGLV5-48*, *UEVLD*, and *CENPBD1P1*) with variations following an autosomal recessive inheritance pattern (Table 2). Additionally, the proband carries two non-inherited autosomal heterozygous genes, three missense variants, two inframe deletions, and five splice region variants (Table 3).

### 2.4. Retrospective Analysis of the Literature on Oral Clinical Features

We retrieved a total of 22 relevant case reports and review articles, encompassing 84 patients. However, no studies in the literature addressing “ Fahr’s disease” with oral manifestations were identified. Additionally, there are two studies pertaining to Fahr’s syndrome, involving two patients. It is important to note that the title of one of the articles in the search results included the term “ Fahr’s disease”; however, after a careful verification, it was determined that the patient should be classified under “ Fahr’s syndrome.” Consequently, this article was included in the relevant literature for the subsequent analysis of Fahr’s syndrome. Nine documents were related to IHP, involving 13 patients, and 11 documents addressed pseudohypoparathyroidism, involving 69 patients. The oral clinical features of these patients, including “tooth agenesis”, “abnormal tooth eruption”, “ dental malformation”, and “root dysplasia” are detailed in Table 4. In addition, we conducted a statistical analysis of the incidence of these characteristics, revealing that the prevalence of “tooth agenesis”, “root dysplasia”, “dental malformation”, and “abnormal tooth eruption” across all cases reported in the literature was 39.29%, 52.39%, 55.95%, and 44.05%, respectively. Among these, their incidence in patients with PHP was 34.78%, 52.17%, 53.62%, and 42.03%, respectively. In contrast, their incidence in patients with IHP was 53.85%, 53.85%, 69.23%, and 46.15%, respectively. These findings indicate a notable difference in the incidence of “ tooth agenesis “ and “ dental malformation” between the two diseases (Table 5).

## 3. Discussion

Basal ganglia calcification can be categorized into two distinct types based on etiology: primary (Fahr’s disease) and secondary (Fahr’s syndrome) [33]. Fahr’s disease predominantly occurs in familial or sporadic cases, often inherited in an autosomal dominant manner, while Fahr’s syndrome is more commonly secondary to underlying conditions such as hypoparathyroidism. Although both conditions share similar clinical manifestations—seizures, movement disorders, cognitive decline, or neuropsychiatric symptoms—there are significant differences in their underlying causes, prognosis, and therapeutic approaches [34]. In this study, the proband exhibited recurrent epileptic seizures, with biochemical analyses indicating classic signs of pseudohypoparathyroidism, including hypocalcemia, hypokalemia, and markedly elevated PTH levels. Notably, CT imaging revealed the hallmark bilateral basal ganglia calcifications commonly associated with both Fahr’s disease and Fahr’s syndrome. However, a further investigation of the proband’s family members showed no clinical or laboratory abnormalities, and the WES failed to detect any pathogenic mutations linked to Fahr’s disease. Taking into account the patient’s medical history, family background, clinical presentation, biochemical findings, radiological evidence, and WES results, we conclude that this case represents Fahr’s syndrome secondary to pseudohypoparathyroidism, rather than primary Fahr’s disease.

Research indicates that the homeostasis of calcium and phosphate metabolism is critical during tooth development and eruption [35]. Consequently, metabolic changes, particularly those associated with abnormal calcium metabolism, are likely to lead to abnormalities in tooth and bone structure. For instance, studies indicate that phosphate transporters, such as *Xpr1*, *Slc20a1*, and *Slc20a2*, are highly expressed in the mouse postnatal tooth germ, and the knockout of *Slc20a2* in mice leads to significant impairments in dentin mineralization [36]. This finding suggests that these transporters may play an indirect role in regulating the availability of extracellular phosphate to mineralized cells and contribute to the regulation of tooth mineralization. However, the changes in oral tissues associated with Fahr’s syndrome remain inadequately documented to date, with only two case reports identifying oral abnormalities in patients with this condition, involving a total of two patients [5,13]. We compared and analyzed the oral clinical features of the proband reported in this study with those of the two previously documented patients. Our findings revealed that all three patients exhibited tooth agenesis and abnormal tooth eruption, while two of the patients displayed clinical characteristics indicative of root dysplasia and dental malformation. This underscores the necessity for oral surgeons to recognize the symptoms associated with Fahr’s syndrome to facilitate appropriate referrals, diagnosis, and treatment.

Currently, there is no standardized dental management for Fahr’s syndrome, largely due to its rarity. Treatment for patients with pseudohypoparathyroidism or similar conditions typically focuses on calcium and vitamin D supplementation to manage metabolic imbalances. Orthodontic treatment can address impacted or unerupted teeth, but clinicians should be cautious of root dysplasia and shortened roots, which increase the risk of root resorption during treatment. For severe dental issues such as enamel hypoplasia or failed tooth eruption, prosthodontic solutions like implants or dentures may be necessary to restore function. PTH is critical not only in the development of Fahr’s syndrome but also in tooth formation and eruption [37]. Both PTH and PTH-related peptide (PTHrP) play key roles in these processes [38], with disturbances potentially affecting root development and tooth eruption. In patients with hypoparathyroid conditions, features like blunted root apices and shortened roots are often observed, as seen in our patient with developmental abnormalities in multiple teeth [24,39]. Research shows that tooth agenesis and abnormal tooth eruption are common in patients with PHP [15], with about 30% of hypoparathyroidism cases exhibiting these dental anomalies [40]. One study found that 29% of PHP patients and 55% of those with IHP experience tooth agenesis or abnormal eruption [24]. This retrospective analysis of 82 patients across 20 PHP and IHP case reports revealed that 34.78% of PHP and 53.85% of IHP patients had tooth agenesis. Additionally, root dysplasia was observed in 52.17% of PHP and 53.85% of IHP patients, while dental malformations affected 53.62% of PHP and 69.23% of IHP patients. Abnormal tooth eruption was reported in 42.03% of PHP and 46.15% of IHP patients. These findings highlight a consistent pattern of oral abnormalities across both conditions, reinforcing earlier studies.

Molecular genetic analysis plays a key role in diagnosing rare hereditary diseases. In this study, WES was conducted on the proband, her parents, and her siblings, focusing on autosomal recessive inheritance, de novo mutations, and compound heterozygous mutations. Despite the thorough analysis, no known pathogenic variants were found in genes typically associated with Fahr’s syndrome (e.g., *XRP1*, *PDGFRB*, *JAM2*, *PDGFB*, *SLC20A2*, or *MYORG*) or pseudohypoparathyroidism (e.g., *GNAS*, *STX16*, or *GNASAS1*). However, several genetic variants in *KIF1A*, *PDGFA*, and *FZD8* were detected, which could underlie the neurological and dental symptoms observed in the patient. KIF1A, part of the kinesin family, is responsible for intracellular transport, particularly in synaptic vesicle movement. Mutations in this gene have been linked to neurodegenerative conditions like Rett syndrome, in which patients exhibit dental issues such as bruxism, anterior open bite, and enamel hypoplasia [41]. These findings are similar to those in our patient, suggesting KIF1A may regulate tooth development, particularly affecting tooth eruption and enamel formation. PDGFA plays a critical role in both bone biology and tooth development (odontogenesis). It interacts with PTH to regulate bone formation [42], but its role extends into early tooth development, influencing tooth size, shape, and ameloblast differentiation—the cells responsible for enamel formation [43]. *Pdgfa* knockout models in animals have shown significant tooth defects, including abnormal tooth size and shape as well as delayed eruption, likely due to its influence on mesenchymal–epithelial signaling during tooth development [44]. FZD8, a receptor in the Wnt signaling pathway, is essential for tooth morphogenesis. It regulates the critical interactions between epithelial and mesenchymal tissues, which shape the enamel knot and guide stem cell activity during tooth formation. Mutations in *FZD8* disrupt this process, leading to dental abnormalities like enamel and dentin defects and delayed tooth eruption [45]. Additionally, *Fzd8* knockout models demonstrate impaired bone remodeling, which can further complicate normal tooth eruption and result in impacted teeth [46]. These findings suggest that variants in *KIF1A*, *PDGFA*, and *FZD8* may collectively contribute to both the neurological and dental manifestations of Fahr’s syndrome, shedding light on the broader genetic mechanisms involved.

Our WES revealed mutations in genes critical to neural and dental development in the proband, supporting a genotype–phenotype link in Fahr’s syndrome. However, due to the limited sample size, these findings remain tentative, highlighting the need for larger studies to clarify the contributions of genetics to the oral and neurological manifestations of the syndrome. Dentists play a vital role in recognizing the dental abnormalities associated with PHP. With clinical, laboratory, and radiological tools, dentists can identify key oral manifestations—such as dental malformations, tooth agenesis, and root dysplasia—that suggest the presence of PHP or IHP. Early detection is critical for timely interventions and can significantly improve long-term outcomes. Orthodontic treatments, in particular, must account for the presence of shortened roots, as these increase the risk of root resorption and complicate the treatment process [47]. Despite the insights provided, this study has limitations. The absence of serial imaging makes it difficult to track the progression of both neurological and dental abnormalities over time. Such longitudinal data would offer a clearer picture of how calcium and phosphate metabolism disorders contribute to the observed dental phenotypes. Moreover, this study’s findings are based on a single proband and retrospective data, limiting its generalizability. Larger cohort studies are needed to validate the suggested genetic links involving *KIF1A*, *FZD8*, and *PDGFA*. Future research should include whole-genome sequencing to identify novel gene variants and combine longitudinal clinical assessments with biochemical and imaging studies. This approach will help clarify how metabolic disturbances affect brain calcifications and dental structures, enabling more targeted therapeutic strategies.

## 4. Methods

### 4.1. Case Presentation

The proband, a 23-year-old female with epilepsy, chronic hepatitis B, and pseudohypoparathyroidism, sought treatment at the Rare Oral Genetic Disease Clinic of the School of Stomatology at the Fourth Military Medical University due to having multiple unerupted teeth. Her family reported that she started having seizures at the age of five, which increased in frequency and duration over time. Approximately six years ago, she received a diagnosis of Fahr’s syndrome at another hospital. Following treatment with calcium carbonate D3 tablets and vitamin D drops, her epilepsy symptoms showed improvement. Notably, her parents and two brothers did not display any similar abnormalities.

### 4.2. Clinical and Biochemical Examination

Basic information was collected from the proband, including her family history, drug allergy history, surgical history, and other relevant details. A routine physical examination was conducted, focusing on the proband’s skin and mucous membranes, head and neck, oral cavity, spine, chest, and abdomen. Blood samples were obtained from the proband, her parents, and siblings and analyzed using an automated biochemical analyzer to assess serum levels of calcium, phosphorus, and parathyroid hormone, with the results compared against reference ranges to identify trends. Additionally, radiological assessments included both computed tomography (CT) and cone-beam computed tomography (CBCT). For the brain CT scans, calcifications were identified based on specific criteria: (1) CT Hounsfield unit (HU) values, with normal brain tissue ranging from 30 to 40 HU, and calcifications typically exceeding 100 HU; and (2) clear demarcation of the calcified regions, with distinct contrast to surrounding tissue. All procedures involving human participants adhered to the ethical standards set forth by the institutional and/or national research committee, as well as the 1964 Helsinki Declaration and its subsequent amendments or comparable ethical standards. Ethical approval was obtained from the Ethics Committee of the Stomatological Hospital, the Fourth Military Medical University (Xi’an, China). 

### 4.3. Pedigree and Gene Analysis

A family consisting of five members participated in the study. The proband (II3), her father (I1), mother (I2), and two older brothers (II1, II2) provided peripheral blood samples for analysis (Figure 4). Genomic DNA (gDNA) was extracted using the QIAamp DNA Blood Mini kit (Qiagen, Valencia, CA, USA) following the manufacturer’s instructions. WES, which included exome capture, high-throughput sequencing, and common filtering, was conducted at the Beijing Genomics Institute (BGI, Shenzhen, China). Sequence reads were aligned, the reference genome was indexed, and variant calling and annotation were performed using the Agilent SureSelect Human All Exon V6 system (Agilent Technologies, Santa Clara, CA, USA). Valid sequencing data from WES were aligned to the human reference genome using the Maq program 2021, with the reference assembly being hg19. Rare variants with a minor allele frequency (MAF) of less than 0.05 were selected from various databases including Exome Aggregation Consortium (ExAC), Exome Variant Server (EVS), 1000 Genomes Project, dbSNP, dbVar, GnomAD, NHLBI GO Exome Sequencing Project, Hapmap, and Scripps Wellderly Genome Resource. Following the 2015 standards of the American Society for Medical Genetics (ACMG), potential gene variants were identified by analyzing different types of variants such as missense, frameshift, inframe insertion, inframe deletion, splice region, splice donor, splice acceptor, stop gained, and stop lost. All variants were validated using ClinVar, OMIM, and HGMD databases.

### 4.4. Literature Collection and Analysis

To further investigate potential similarities in the oral clinical features between the cases in this study and other reported cases, we employed the terms “Fahr’s disease”, “Fahr’s syndrome”, “dental”, “oral manifestation”, and “hyperparathyroidism”. These were combined with various search strategies across platforms such as PubMed, Google Scholar, OMIM, and other relevant websites to retrieve pertinent case reports and studies. The literature search encompassed a time range from 1 January 1965 to 30 June 2024. Specifically, we meticulously filtered the literature according to the following stringent inclusion criteria: (1) studies that provided comprehensive intraoral photographs, allowing for precise evaluation of oral manifestations; (2) studies that included high-resolution imaging, such as X-rays, panoramic tomographs, or CT scans; and (3) in cases for which visual documentation was absent, detailed descriptions of oral anomalies, including the number and location of missing or impacted teeth, were required to facilitate accurate data analysis. Any studies failing to meet these criteria were excluded from further consideration. Our analysis focused on common oral clinical features associated with these conditions, specifically regarding “tooth agenesis”, “root dysplasia”, “dental malformation”, and “abnormal tooth eruption”. Furthermore, we conducted statistical comparisons of the incidence rates of the aforementioned oral features.

## Figures and Tables

**Figure 1 ijms-25-11611-f001:**
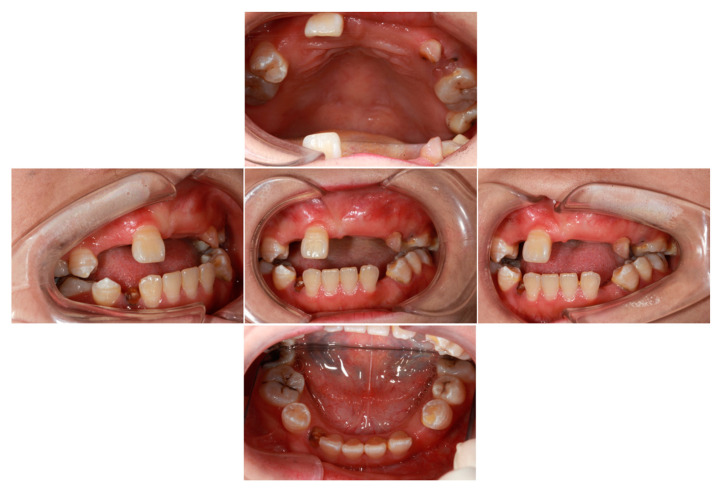
Oral examination of the proband revealed multiple unerupted teeth. The left upper deciduous canine and the right lower deciduous canine (root remnant) were retained. Additionally, enamel hypoplasia was observed in the right upper second premolar, right lower first premolar, and left lower first premolar.

**Figure 2 ijms-25-11611-f002:**
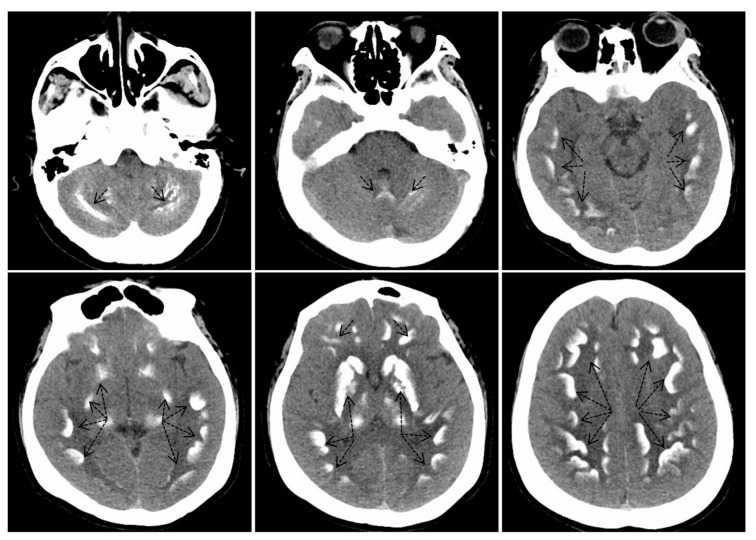
Noncontrast computed tomography images of the head revealed calcifications in the bilateral basal ganglia, encompassing the caudate nucleus, putamen, globus pallidus, thalamus, and dentate nuclei. The black dashed arrows in the figure indicate areas of calcification.

**Figure 3 ijms-25-11611-f003:**
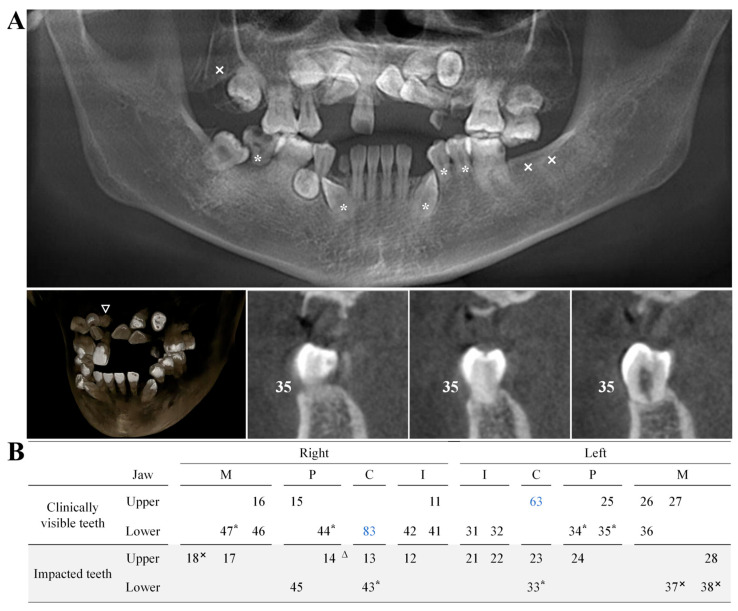
(**A**) CBCT imaging of the proband revealed multiple impacted teeth, with the absence of the lower right second molar (marked with “×” in the figure). Additionally, a supernumerary tooth was identified on the palatal side between the upper right canine and the first premolar (denoted by “△”). (**B**) Developmental anomalies were observed in the roots of the lower left canine, first and second premolars, and in the lower right canine, first premolar, and second molar (indicated by “*”). Retained deciduous teeth are marked in blue, corresponding to the upper left deciduous canine and the residual root of the lower right deciduous canine.

**Figure 4 ijms-25-11611-f004:**
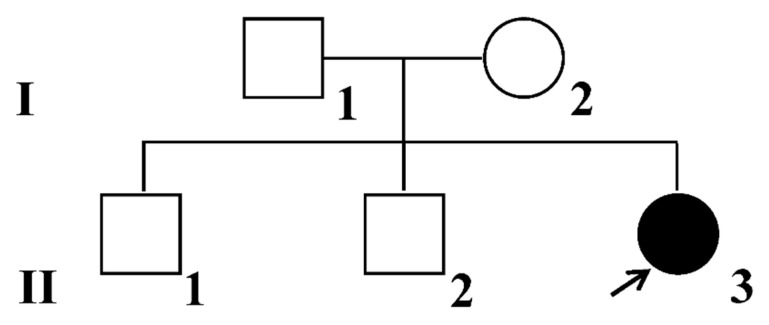
The family genealogy diagram of this study includes the proband’s father (I1), mother (I2), two older brothers (II1, II2), and the proband (II3).

**Table 1 ijms-25-11611-t001:** Laboratory results of the proband.

Items	Values in the Patient	Reference Values
Calcium (mmol/L)	**1.63↓**	2.1–2.5
Sodium (mmol/L)	138	137–147
Potassium (mmol/L)	**2.91↓**	3.5–5.3
Chloride (mmol/L)	**96↓**	99–110
Free triiodothyronine (FT3) (pmol/L)	5.2	3.85–6.30
Free thyroxine (FT4) (pmol/L)	18.1	12.8–21.3
Thyroid-stimulating hormone (TSH) (ulU/mL)	0.71	0.75–5.60
Parathyroid hormone (ng/L)	**412.7↑**	15–65

**↑** denotes that the indicator exceeds the standard reference range; **↓** indicates that it falls below the standard reference range.

**Table 2 ijms-25-11611-t002:** Pathogenic genes fit autosomal recessive inheritance pattern in the transmitted ways.

Gene	CHR	mRNA	Protein	Variation Type	ACMG Judgement	Deleteriousness Assessment	Annotation
*KIF1A*	2	ENST00000404283.9 (c.2751_2753del)	p. Glu917del	In frame deletion	Likely pathogenic	Moderate	neurodegenerative diseases
*UEVLD*	11	ENST00000300038.7 (c.613-4_613-3del)	/	Splice region variant	Likely pathogenic	Low	carbohydrate metabolic process
*OR2T35*	1	ENST00000641268.1 (c.956_957dup)	p. Ile320Ter	Frameshift variant	Uncertain significance	Moderate	olfactory transduction
*IGLV5-48*	22	ENST00000390293.1 (c.185G>A)	p. Gly62Glu	Missense variant	Uncertain significance	Moderate	autoimmune disease
*IGLV5-48*	22	ENST00000390293.1 (c.209A>G)	p. Asn70Ser	Missense variant	Uncertain significance	Moderate	autoimmune disease
*IGLV5-48*	22	ENST00000390293.1 (c.47-5del)	/	Missense variant	Uncertain significance	Low	autoimmune disease
*CENPBD1P1*	19	ENST00000487264.5 (n.347-5_347-3dup)	/	Splice region variant	Uncertain significance	Low	/
*CENPBD1P1*	19	ENST00000487264.5 (n.347-8G>T)	/	Splice region variant	Likely pathogenic	Low	/

CHR: chromosome. “/” indicates absence.

**Table 3 ijms-25-11611-t003:** De novo variants in novel pathogenic genes.

CHR	Genotype	Gene	mRNA	Protein	Annotation
**Missense Variants**					
chr4	Heterozygous	*TRIML2*	ENST00000326754.7: c.1035G>T	p. Arg345Ser	Alzheimer’s disease
chr7	Heterozygous	*SMARCD3*	ENST00000262188.13: c.218C>T	p. Ala73Val	neurodegenerative diseases
chr18	Heterozygous	*TUBB8B*	XM_024451143.1: c.11C>T	p. Pro4Leu	neurodevelopmental disorders and degenerative diseases
**Inframe Deletion**					
chr2	Heterozygous	*FAM117B*	ENST00000392238.3: c.246_248del	p. Gly84del	gastric cancer; familial amyotrophic lateral sclerosis
chr10	Heterozygous	*FZD8*	ENST00000374694.3: c.1929_1946del	p. Pro644_Gly649del	neurodegenerative diseases, developmental disorders, and skeletal disorders
**Splice region Variants**					
chr3	Heterozygous	*KPNA4*	ENST00000334256.9: c.115-7_115-4dup	/	neurodegenerative diseases
chr7	Heterozygous	*PDGFA*	XM_011515415.1: c.72+3G>T	/	cardiovascular disease, dental development
chr7	Heterozygous	*TRIM50*	XM_011515787.1: c.875-5_875-2dup	/	neurological disorders, immune regulation
chr11	Heterozygous	*ANKRD42*	ENST00000531869.1: n.91-3del	/	cardiovascular disease, cancer
chr13	Heterozygous	*ABCC4*	ENST00000642524.1: c.1-5_1-4dup	/	kidney disease

**Table 4 ijms-25-11611-t004:** Retrospective analysis of the literature on oral clinical features.

Case Number	Dental Manifestations *				Clinical Diagnosis	References
	Tooth Agenesis	Root Dysplasia	Dental Malformation	Abnormal Tooth Eruption		
1	1	1	1	1	Fahr’s syndrome	Our case
1	1	0	1	1	Fahr’s syndrome	[5]
1	1	1	0	1	Fahr’s syndrome	[13]
29	/	16	8	/	PHP	[14]
4	4	3	4	2	PHP	[15]
1	0	0	0	1	PHP	[16]
1	1	1	1	1	PHP	[17]
19	10	6	11	14	PHP	[9]
4	2	3	4	3	PHP	[18]
1	0	1	1	0	PHP	[19]
1	1	1	1	1	PHP	[20]
1	1	0	0	1	PHP	[21]
1	0	1	1	0	PHP	[22]
1	1	0	1	1	PHP	[23]
6	4	4	5	5	PHP	[24]
5	3	1	1	2	IHP	[24]
1	1	0	1	0	IHP	[25]
1	1	1	1	1	IHP	[26]
1	0	1	1	0	IHP	[27]
1	0	1	1	1	IHP	[28]
1	0	0	1	0	IHP	[29]
1	0	0	1	0	IHP	[30]
1	1	1	1	1	IHP	[31]
1	1	1	1	1	IHP	[32]
85	34	45	48	38	N/A	Summary

‘PHP’ represents pseudohypoparathyroidism, ‘IHP’ represents idiopathic hypoparathyroidism, ‘/’ represents negative or unknown. ‘*’ represents the number of patients exhibiting the corresponding oral manifestations.

**Table 5 ijms-25-11611-t005:** Statistical analysis of oral clinical features of PHP and IHP patients.

Case Number	Dental Manifestations				Clinical Diagnosis
	Tooth Agenesis	Root Dysplasia	Dental Malformation	Abnormal Tooth Eruption	
69	24/69 (34.78%)	36/69 (52.17%)	37/69 (53.62%)	29/69 (42.03%)	PHP
13	7/13 (53.85%)	7/13 (53.85%)	9/13 (69.23%)	6/13 (46.15%)	IHP

‘PHP’ represents pseudohypoparathyroidism; ‘IHP’ represents idiopathic hypoparathyroidism.

## Data Availability

The clinical data supporting the findings of this study are available upon request from the corresponding author. These data are not publicly accessible due to privacy and ethical restrictions.

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
