# Peer review of "Fahr’s Syndrome with Pseudohypoparathyroidism: Oral Features and Genetic Insights"

_ijms, 2024, doi:10.3390/ijms252111611_

Round 1

Reviewer 1 Report

Comments and Suggestions for Authors

Title: The title is clear and matches with the details presented in the manuscript.

Abstract: The abstract explores a critical and unexplored manifestations of oral manifestations in Fahr’s syndrome. The abstract is well written and clear. The analysis of the oral manifestations is clear, but a further explanation is needed to explain how the oral abnormalities are different between idiopathic hypoparathyroidism and pseudohypoparathyroidism which would further enhance the strength of the study.

Introduction:

The introduction is clear and provides a clear view of differentiation between the two conditions: Fahr syndrome and Fahr disease. The focus on parathyroid hormonal abnormalities associated with oral manifestations in Fahr syndrome is very promising. However, the introduction would improve with a clearer focus on the lack of understanding regarding oral symptoms. Although it notes the presence of oral characteristics, providing more detail regarding the importance of these observations in terms of both clinical application and existing research would enhance the justification for the study. Moreover, including citations to important sources or research on oral symptoms would provide additional information.

Method:

Method applied is very clear and understandable. However, additional information can be added to the methodology by providing some more information about the criteria used to select patients and family members for the study, and explaining the reasoning behind the chosen diagnostic tests, specifically regarding oral symptoms. Moreover, there is no explanation provided about the statistical techniques that was used to evaluate the biochemical and radiological information. Expanding on the informed consent process, particularly due to the proband's family participation, would enhance the already positive ethical adherence section. Providing further explanation on the management and analysis of data would increase the strength of the methods.

Discussion:
The discussion section is well written. However deeper explanation focusing on the practical significance of the result would improve the discussion part. Although a potential regulatory functions of KIF1A, PDGFA, and FZD8 in dental development is clear, there is no discussion regarding the functional mechanisms through which these genes could overall impact the oral characteristics that has been observed.  Moreover, it is also very important to compare the similar genetic findings in other studies to strengthen the genetic correlation in the case presented in this study.

Author Response

Comments 1: The title is clear and matches with the details presented in the manuscript.

Response 1: Dear reviewer, thank you for reviewing our manuscript and for the constructive comments, which greatly helped us to improve the manuscript. The manuscript was carefully revised and point-by-point response was listed below. We hope that your comments have been addressed accurately.

Comments 2: The abstract explores a critical and unexplored manifestations of oral manifestations in Fahr’s syndrome. The abstract is well written and clear. The analysis of the oral manifestations is clear, but a further explanation is needed to explain how the oral abnormalities are different between idiopathic hypoparathyroidism and pseudohypoparathyroidism which would further enhance the strength of the study.

Response 2: Dear Reviewer, thanks for your valuable comments and suggestions on our manuscript. In response, we have made corresponding revisions to the abstract. The revised abstract now briefly outlines the similarities and differences in oral abnormalities, such as tooth agenesis and dental malformation, observed in patients with Fahr's syndrome, idiopathic hypoparathyroidism, and pseudohypoparathyroidism. All revisions are highlighted in red and can be found in the “Abstract” section on the first page of the revised manuscript. However, due to the 200-character limit for the abstract, more detailed explanations are provided in the "Results" section of the revised manuscript. Once again, we appreciate your insightful suggestions.

Comments 3: The introduction is clear and provides a clear view of differentiation between the two conditions: Fahr syndrome and Fahr disease. The focus on parathyroid hormonal abnormalities associated with oral manifestations in Fahr syndrome is very promising. However, the introduction would improve with a clearer focus on the lack of understanding regarding oral symptoms. Although it notes the presence of oral characteristics, providing more detail regarding the importance of these observations in terms of both clinical application and existing research would enhance the justification for the study. Moreover, including citations to important sources or research on oral symptoms would provide additional information.

Response 3: Dear Reviewer, thank you very much for your thorough review and constructive comments on our manuscript. We have incorporated your suggestions and revised the "Introduction" section accordingly. The detailed revisions can be found in the "Introduction" section on pages 1 to 2 of the revised manuscript. All changes have been made in track-change mode and highlighted in red for easy reference.

Comments 4: Method applied is very clear and understandable. However, additional information can be added to the methodology by providing some more information about the criteria used to select patients and family members for the study, and explaining the reasoning behind the chosen diagnostic tests, specifically regarding oral symptoms. Moreover, there is no explanation provided about the statistical techniques that was used to evaluate the biochemical and radiological information. Expanding on the informed consent process, particularly due to the proband's family participation, would enhance the already positive ethical adherence section. Providing further explanation on the management and analysis of data would increase the strength of the methods.

Response 4: Dear Reviewer, we sincerely appreciate your thorough review and constructive feedback. In response to your suggestions, we have now implemented revisions, all of which are clearly marked in red within the revised manuscript for your ease of review. In the revised manuscript, we have added further details regarding the biochemical analyses and radiological examinations in the “2.2. Clinical and biochemical examination” section under "Results." Additionally, we included a statement regarding the ethical approval granted for the proband and her family members' participation in the study. These changes can be found at the end of page 2 and the beginning of page 3 in the revised manuscript. Moreover, in the "2.4. Literature collection and analysis" section of the "Results," we have clarified the specific methods used for screening and analyzing the literature. We are confident that these revisions provide the clarity and precision you requested. The detailed content can be found at the end of page 3 and the beginning of page 4 in the revised manuscript.

Comments 5: The discussion section is well written. However deeper explanation focusing on the practical significance of the result would improve the discussion part. Although a potential regulatory functions of KIF1A, PDGFA, and FZD8 in dental development is clear, there is no discussion regarding the functional mechanisms through which these genes could overall impact the oral characteristics that has been observed. Moreover, it is also very important to compare the similar genetic findings in other studies to strengthen the genetic correlation in the case presented in this study.

Response 5: Dear Reviewer, we sincerely appreciate your constructive feedback on the discussion section of our manuscript. In response to your insightful suggestions, we have strengthened the practical implications of our findings and expanded the discussion on the potential mechanisms by which KIF1A, PDGFA, and FZD8 might influence the observed dental abnormalities. Despite our efforts, we were unable to find comparable case reports in the existing literature that align closely with our genetic findings. Nevertheless, we hope these revisions address your concerns and provide additional depth to our discussion on genotype-phenotype correlations. You can find the detailed revisions at the end of page 10 and the beginning of page 11 in the revised manuscript.

Reviewer 2 Report

Comments and Suggestions for Authors

The authors have collected significant amount of data, which is presented in the manuscript. Nonetheless, the reviewer considers that is vital to address the following, to the publication be endorsed:

1. The reviewer major concern relates to the aim/methods of the current study. Authors have chosen to focus on the oral manifestations of Fhar´s syndrome secundary pseudohypoparathyroidism, in which clinical manifestations are consequence of metabolic disruption, which is acknowledged by the authors in the introduction section. In this sense, it is not clear why authors have conducted WES in the proband of the clinical case. A detailed rationale must be presented to justify that, in order to assess its relevance and the relevance of its findings, that were further discussed in the discussion section.

2. Authors purpose to describe the relation beteween Fahr´s syndrome and impaired oral/tooth developement, by performing an extensive review of the literarture. The results show that the majority of cases, presented by the authors, refer to IHP or PHP cases. It is not clear if those cases have indeed a Fahrh´s syndrome diagnosis or limited to the  endocrine disturbance. In this sense, it is not clear what is the aim of the authors. Authors may redefine the aim of the sudy, in accordance.

3. Figure 3 must have clear landmarks of relevant anatomical structures and imagiological findings, with its respective legend on the caption, similar to Figure 4.

Author Response

Comments 1: The reviewer major concern relates to the aim/methods of the current study. Authors have chosen to focus on the oral manifestations of Fhar’s syndrome secondary pseudohypoparathyroidism, in which clinical manifestations are consequence of metabolic disruption, which is acknowledged by the authors in the introduction section. In this sense, it is not clear why authors have conducted WES in the proband of the clinical case. A detailed rationale must be presented to justify that, in order to assess its relevance and the relevance of its findings, that were further discussed in the discussion section.

Response 1: Dear Reviewer, thanks for your invaluable feedback. The distinction between Fahr's syndrome and Fahr's disease remains a subject of ongoing debate. While these terms are often used interchangeably in the literature, some scholars argue they represent distinct conditions. Both disorders are linked to genetic factors, but Fahr's syndrome typically has a more complex etiology, frequently associated with secondary conditions like hypoparathyroidism, whereas Fahr's disease tends to be idiopathic in nature. Our patient exhibited recurrent seizures alongside biochemical indicators of pseudohypoparathyroidism, as well as radiographic evidence of symmetric calcification in the basal ganglia, including the putamen, caudate nucleus, and thalamus—hallmark features of both Fahr's syndrome and Fahr's disease. These overlapping clinical characteristics posed significant diagnostic challenges. To further elucidate the genetic background and facilitate a more precise molecular diagnosis, we performed whole-exome sequencing on the patient and her family members. While no pathogenic mutations were detected in genes typically associated with Fahr's disease, based on the clinical presentation, family history, laboratory results, and radiological findings, we concluded that this case is more consistent with Fahr's syndrome secondary to pseudohypoparathyroidism, rather than Fahr's disease. This discussion has been expanded in the revised manuscript, with all modifications highlighted in red. The relevant updates can be found at the end of page 9 and the beginning of page 10. Once again, we greatly appreciate your insightful feedback.

Comments 2: Authors purpose to describe the relation between Fahr’s syndrome and impaired oral/tooth development, by performing an extensive review of the literature. The results show that the majority of cases, presented by the authors, refer to IHP or PHP cases. It is not clear if those cases have indeed a Fhar’s syndrome diagnosis or limited to the endocrine disturbance. In this sense, it is not clear what is the aim of the authors. Authors may redefine the aim of the study, in accordance.

Response 2:  Dear Reviewer, we sincerely appreciate your insightful comments. We apologize for any misunderstanding regarding the true objective of our study in the previous version of the manuscript. I would like to provide a detailed explanation here. Our research initially focused on the clinical phenotypes and genetic background of a patient with Fhar’s syndrome accompanied by pseudohypoparathyroidism, as well as the patient’s family members. Given that Fhar’s syndrome is most commonly secondary to parathyroid-related disorders, and considering that our patient was diagnosed with pseudohypoparathyroidism, we subsequently conducted a systematic review and comparison of the oral abnormalities in this patient with those of other Fhar’s syndrome patients, as well as patients with isolated idiopathic hypoparathyroidism or isolated pseudohypoparathyroidism. Previous knowledge on this aspect has been limited, but through our study, we found that these patients exhibit relatively similar oral abnormalities, which we hypothesize may be closely linked to the endocrine disturbances associated with these conditions.

Comments 3: Figure 3 must have clear landmarks of relevant anatomical structures and imagiological findings, with its respective legend on the caption, similar to Figure 4.

Response 3: Dear Reviewer, we sincerely appreciate your insightful suggestions. In response, we have revised Figure 3 by distinctly annotating the characteristic calcifications in the brain using black dashed arrows, as recommended. Additionally, we have updated the figure legend accordingly, with the revisions highlighted in red.

Round 2

Reviewer 2 Report

Comments and Suggestions for Authors

Authors have succeed in addressing the reviewer's concerns.